# Deep Joint Spatiotemporal Network (DJSTN) for Efficient Facial Expression Recognition

**DOI:** 10.3390/s20071936

**Published:** 2020-03-30

**Authors:** Dami Jeong, Byung-Gyu Kim, Suh-Yeon Dong

**Affiliations:** Department of IT Engineering, Sookmyung Women’s University, 100 Chungpa-ro 47 gil, Yongsna-gu, Seoul 04310, Korea; dm.jeong@ivpl.sookmyung.ac.kr (D.J.); sydong@sookmyung.ac.kr (S.-Y.D.)

**Keywords:** facial expression recognition (FER), deep learning, local binary pattern (LBP) feature, geometric feature, deep spatiotemporal network, joint fusion classifier

## Abstract

Understanding a person’s feelings is a very important process for the affective computing. People express their emotions in various ways. Among them, facial expression is the most effective way to present human emotional status. We propose efficient deep joint spatiotemporal features for facial expression recognition based on the deep appearance and geometric neural networks. We apply three-dimensional (3D) convolution to extract spatial and temporal features at the same time. For the geometric network, 23 dominant facial landmarks are selected to express the movement of facial muscle through the analysis of energy distribution of whole facial landmarks.We combine these features by the designed joint fusion classifier to complement each other. From the experimental results, we verify the recognition accuracy of 99.21%, 87.88%, and 91.83% for CK+, MMI, and FERA datasets, respectively. Through the comparative analysis, we show that the proposed scheme is able to improve the recognition accuracy by 4% at least.

## 1. Introduction

Affective computing is the study of recognizing human feelings [1]. Using this technology, we can understand people more effectively. People use several types of signals such as audio and visual signal to express their emotion [2]. In common with voice, tone, and gesture, facial expressions are an important means to express emotional state. Facial expression recognition is a technique that automatically extracts the features on human face to recognize the patterns of expression. In general, it identifies the state of emotions by classifying 6–8 major emotions such as angry, disgust, fear, happy, sad, and surprise.

Besides these eight emotions, there are also many types of research for classifying subtle expression. To reach this goal, it should be able to analyze the facial expression even if it is not a typical expression. Facial Action Coding System (FACS) can be a tool to understand the human face [3]. This is a system for classifying human facial movements by their appearance on the face. FACS consists of facial Action Units (AUs) and it represents facial expression using numbers and alphabets. This system divides facial expression from trace to maximum. In addition, there is much research being undertaken on understanding natural emotion of human. For example, Affectiva MIT Facial Expression Dataset (AMFED) [4] is one that captures the user’s facial expression when watching online media and Acted Facial Expressions in the Wild (AFEW) [5] is created by capturing scenes where actors’ emotions are revealed in the movie clips. Moreover, there is also a dataset to classify the natural facial expressions when people have shoulder pain [6].

Recently, the importance for data increases consistently with the development of deep learning technologies. For training a deep learning model successfully, sufficient and high-quality data are required, since many high-quality data allow better performance, regardless of the size or depth of network. Therefore, data augmentation is required to increase the amount of data. In this way, the accuracy of facial expression recognition is increasing to near perfection. However, there is another problem, namely dependency on dataset [7,8]. This is because other factors except facial expressions have a large effect on appearance. Several existing facial expression datasets cover different gender, races, and ages. In addition, external factors such as lighting, pose, resolution, noise, etc. make it highly variable even within the same database. Datasets must be rich enough to accept all these variants for facial expression recognition. To overcome these problems, researchers make cross-dataset that use multiple datasets for learning [9]. Mayer et al. [10] and Hasani et al. [11] experimented with different datasets using one network under the same conditions. Through these experiments, generality of the network can be seen. Figure 1 shows the external differences seen in the facial expression dataset.

The temporal dynamic problem is another challenge, although the accuracy of facial expression is increasing to near perfection due to these efforts. Many facial expression datasets consist of images captured under controlled experimental conditions. For example, most were shot in frontal pose to avoid shadows. In this way, datasets taken in laboratory environments have controlled lighting, poses, and brightness. However, these constraints are not controlled in the wild environment. For actual application, temporal dynamics should be considered.

We propose a new deep joint spatiotemporal network (DJSTN) to solve these problems. First, we use 3D CNN structure that can extract spatial feature and temporal feature at the same time to get more accurate result of facial expression recognition. The combination of multiple frames is used as input to the proposed network. In addition, we combine two networks called the appearance network, which uses facial appearance features, and the geometric network, which used facial landmark information for multiple datasets. Through designing the joint function, two networks are properly combined to improve the performance of the recognition accuracy.

The rest of this paper is organized as follows. The related works for facial expression recognition are introduced in Section 2. Section 3 introduces four main modules required in our overall facial expression recognition system including the employed preprocessing and datasets. Section 4 will provide several experimental results and the performance comparison with state-of-the-art models. Finally, we will make a conclusion of this paper in Section 5.

## 2. Related Works

### 2.1. Facial Expression Recognition Approaches

#### 2.1.1. Classical Approaches

Facial expression recognition is a kind of pattern recognition problem. Thus, there have been many studies on facial expression recognition using classical handcraft feature-based method. For example, Histogram of Gradient (HoG) [12], Local Binary Pattern (LBP) [13], Scale Invariant Feature Transform (SIFT) [14] and Gavor wavelet [15] are representative features. In addition, several machine learning based classifiers have been utilized to classify facial expression. Support Vector Machine (SVM) [16] is one well-known classifier, which considers the decision boundaries with maximum margin.

Vasanth et al. [17] and Abdulrahman et al. [18] proposed classical facial expression recognition algorithms using SVM and other features. Xiaoming et al. achieved better performance in facial expression recognition by using Gabor wavelet feature and HOG feature with SVM classifier [19]. The classical features can expand to 3D HOG [20], LBP-TOP [21], and 3D-SIFT [22] to apply multiple frames. The expanded features were used to recognize facial expression on 3D face database. In [23], the authors proposed the set of selected SIFT feature to improve the performance on 3D face database. Ashish Tawari et al. proposed a new facial expression recognition framework using audio-visual information analysis. They proposed to model the cross-modality data correlation while allowing them to be treated as asynchronous streams [24]. In this method, the SVM is employed as classifier. This approach achieves promising results: about 85% accuracy for binary-class classification for all 15 possible combinations over six basic emotions was achieved. In [25], Kirana et al. used facial-based features to detect face and recognize emotion. They applied rectangular feature and cascading AdaBoost algorithm, which is the main concept of the Viola–Jones algorithm in both processes. This method resulted in 74% recognition accuracy, but operated in real time.

However, for huge datasets such as FER2013 [26] and wild and dynamic environmental datasets such as Acted Facial Expressions In The Wild (AFEW) [27], the accuracy of facial expression algorithms cannot be improved with simple feature extraction. For FER2013 dataset [26], the data consist of 48 × 48 pixel grayscale images of faces. The faces have been automatically registered so that the face is more or less centered and occupies about the same amount of space in each image. This dataset includes many images of characters which have been made graphically. Thus, each image of this dataset is small and there are wide variations of face images, including many images of graphically made characters and of people wearing accessories such as glasses. This makes it difficult to extract the exact features as usual. This may cause the degradation of recognition accuracy.

The AFEW dataset [27] is known as one dynamic temporal facial expressions data corpuses consisting of close to real world environment images extracted from movies. Thus, illumination condition, view of face (not exact frontal view), and size of face image are not controlled when compared to earlier datasets such as JAFFE and CK. Although we can detect the face area from the background, the detected face may not be the exact frontal-view image in many cases. Therefore, it is very difficult to recognize the facial expression with a partial view of face.

#### 2.1.2. Deep Learning-Based Approaches

Recently, most facial expression recognition algorithms use deep learning-based methods. Xiangyun et al. proposed the peak-piloted deep network to train expression intensity-invariance [28]. It takes a pair of peak and non-peak expression images as input and makes that non-peak expression close to the peak expression in the peak gradient suppression step. The hierarchical network structure are also considered as ways to increase the accuracy of facial expression recognition [29].

In addition, some studies using temporal information have been conducted. The 3D CNN method was used for action recognition problem for the first time. Then, it has been applied to facial expression recognition with deformable action part constraints [30]. Strong spatial structural constraints of the dynamic action parts were used based on 3D CNN. In [31], STM-Expressionlet is proposed, which uses three multiple frames as input. Hasani et al. proposed the 3D Inception-Resnet model [11], which was modified from 2D Inception-ResNet model [32] to three-dimensional structure. The combination of multiple structural input and 3D CNN which can support temporal feature extraction greatly improves the accuracy on facial expression recognition problem.

Hybrid schemes using multiple networks has also been used to increase the accuracy of facial expression recognition. In [33], two deep networks called deep temporal appearance network and deep temporal geometric network extracted temporal features from image sequences and facial landmark points. The authors combined these two networks with a new integration method to make the two models cooperate each other. Yin et al. won the EmotiW 2016 challenge [34] with superior performance through the combination of three modules: CNN-RNN, 3D convolution, and SVM with linear kernel [35].

Y. Tian et al. suggested the Secondary information aware Facial Expression Network (SIFE-Net) to explore the latent components without auxiliary labeling and a novel dynamic weighting strategy to teach the SIFE-Net [36]. In this method, another trained network is needed to obtain the secondary information. A. Raheel et al. employed brain signals of individuals to recognize their facial expressions [37]. Subjects were asked to express their facial expressions while watching a video clip and EEG data were recorded using a 14-channel Emotiv/EPOC EEG headset. This kind of approach needs an EEG signal capture device (headset). This causes discomfort. In addition, deep comprehensive multipatches aggregation convolutional neural networks (CNNs) has been reported to solve the FER problem [38]. The proposed method mainly consists of two branches of CNN. One branch extracts local features from image patches while the other extracts holistic features from the whole expressional image. The authors combined them to classify the emotion using very static datasets such as CK+ and JAFFE. Wang et al. proposed a bimodal fusion algorithm to realize speech emotion recognition, where both facial expression and speech information are optimally fused. This method focuses on achieving speech emotion recognition [39]. In [40], Mingjing Yu et al. designed a multi-task learning framework for global-local representation of facial expressions. First, a shared shallow module is designed to learn information from local regions and the global image. Then, they constructed a part-based module, which processes critical local regions including the eyes, nose, and mouth to extract local informative dynamics related to facial expressions. In addition, a global face module was proposed to extract global appearance features related to expressions. To extract local features of the eyes, nose, and mouth, they employed complex network structure comparing to other methods.

In addition, there are some systems for business applications to provide face recognition and emotion recognition technologies [41,42,43,44,45,46,47,48,49,50,51,52,53,54], as shown in Table 1. We summarize the properties of the existing facial emotion recognition systems with various signals. Most of them employ facial expression recognition based on deep learning to make a business model based on customer’s experience.

We also utilize deep learning technique to recognize facial expression more accurately. The proposed method is based on 3D CNNs and the hybrid scheme. The contributions of this study are as follows:We design a new hybrid network structure based on appearance feature (3D CNN) and geometric feature (2D CNN).We experimentally identify dominant facial landmarks with large variation (motion energy) when facial expression changes, to design the geometric feature-based network. We obtain similar effects as when using all landmarks by using only 21 landmarks.We design the joint fusion classifier combining two networks from the appearance feature-based network and geometric feature-based network.

## 3. Proposed Scheme

The overall structure of the proposed method is shown in Figure 2. We provide a detailed description of the proposed scheme below.

### 3.1. Datasets and Data Augmentation

We used three types of datasets for the experiment. The Extended CohnKanade (CK+) dataset is one of most extensively used facial expression databases [55]. It is laboratory-controlled dataset that includes seven expressions: anger, contempt, disgust, fear, happiness, sadness, and surprise. The CK+ dataset contains 593 video sequences from 123 subjects aged 18–30 years. The image resolutions are 640 × 480 and 640 × 490. The length of sequence varies from 10 to 60 frames. Each sequence shows an expression shift from neutral expression to peak expression. Since the CK+ dataset is based on video sequence, we can extract temporal information by using this dataset.

The second is the MMI dataset.This is a laboratory-controlled video and image dataset [56]. It includes 326 videos and high-resolution still images of 75 subjects with 720 × 576 image resolution. Its subjects are both male and female and aged from 19 to 62 years. In the MMI dataset, 213 sequences are labeled with the same seven expressions as the CK+ dataset. Each of the sequence begins with a neutral expression and reaches peak expression near the middle of sequence. It returns to neural expression when the sequence is over. Moreover, the MMI dataset is more challenging because many of the subjects wear accessories such as glasses and moustaches.

The last one is the GEMEP-FERA dataset. The GEMEP-FERA (FERA) [57] dataset is a subset of the Geneva Multimodal Expression Corpus for Experimental Research on Emotion Perception (GEMEP) used as database for the Facial Expression Recognition and Analysis (FERA) challenge [58]. It contains 87 image sequences of seven subjects including four females and three males. In addition, it has five emotion categories: anger, fear, joy, relief, and sadness. The relief expression is not a common label in facial expression recognition task. Joy expression is shown externally. The FERA dataset is also a laboratory-controlled dataset similar to most datasets, but the movement of each subject is faster. In addition, the head pose and subject’s mouth change dynamically because it is filmed when each subject expresses their emotion with talking.

Based on these datasets, we applied data augmentation techniques to increase the amount of data and make input data to reflect various situations. In addition, each dataset has different distributions and numbers of sequences. In machine learning and deep learning for classification problem, it is necessary to have enough data for each class. In addition, it is recommended to have a similar number of data for all classes to provide sufficiently distributed information for the designed network (training) models. Based on this, we needed to make enough and similar number of samples in each dataset. We constructed facial expression input datasets in the following ways: First, we extracted a few frames in which the expression is well revealed from input sequence. If the database has a neutral labeled sequence separately, then neutral labeled data were constructed in the same way. However, since the databases we use do not explicitly indicate neutral label, the first few frames without facial expression were set as neutral data. After extracting frames according to their emotions, each frame was connected to combine them.

For the CK+ dataset, there is no neutral labeled sequence. Thus, we designated the first few frames of each sequence as the neutral label and last few consecutive frames were set to each emotion label. Therefore, the difference of distribution is very large because neutral labeled data can exist in every sequence while the emotion labeled data only can be created from certain labeled sequence. The total number of emotional labeled input data of the CK+ dataset is 300. Therefore, we needed to increase emotion labeled data to avoid over-fitting problem that can occur due to lack of data and biased input data distribution.

To increase the amount of data in each dataset, we applied data augmentation techniques. Horizontal flip was applied on whole datasets that needed to be augmented. Each image was also rotated by each angle in {−7.5${}^{{}^{\circ}}$, −5${}^{{}^{\circ}}$, −2.5${}^{{}^{\circ}}$, 2.5${}^{{}^{\circ}}$, 5${}^{{}^{\circ}}$, 7.5${}^{{}^{\circ}}$}. Through this process, we obtained eight times more data, except the neutral labeled data. Moreover, the CK+ dataset contains contempt labeled images, but the other datasets do not. Therefore, we excluded images that are labeled contempt for setting the same experimental condition. The MMI dataset also needed to be augmented because their total number of input data are not enough. We increased input data from the MMI dataset so that the number of each emotion label data was similar to neutral labeled data of the CK+ dataset. For data augmentation of the MMI dataset, horizontal flip and rotation by each angle in {−5${}^{{}^{\circ}}$, −2.5${}^{{}^{\circ}}$, 2.5${}^{{}^{\circ}}$, 5${}^{{}^{\circ}}$} were applied to obtain five times more data.

Table 2 represents the specific figures of increased data for all datasets. For the FERA dataset, it only has five emotion class and does not have neutral emotion but does have relief emotion. Thus, we referred to relief labeled frames as neutral for experimental setting. The symbol “-” in the table means that the FERA dataset does not have the appropriate label.

### 3.2. Appearance Feature-based Spatiotemporal Network

#### 3.2.1. Feature Extraction

Because resolution and other external features of each dataset used in experiments are various, a preprocessing step was needed to ensure that they have common features. First, Multi-task Cascaded Convolutional Networks (MTCNN) module detects face areas from the input images [59]. Second, it crops only facial area from the entire image to make them the same size and not affected by unnecessary parts such as hair or accessories. Then, the cropped face images are resized to a size of 128 × 128.

Then, features are extracted to reduce computational complexity compared to the original image and emphasize facial features. There are well-known binary features such as local binary pattern (LBP) [60], Binary Robust Invariant Scalable Keypoints (BRISK) [61], Binary Robust Independent Elementary Features (BRIEF) [62], and the Oriented FAST and Rotated BRIEF (ORB) [63]. Among them, the LBP is widely utilized in computer vision, because it is a computationally simple non-parametric texture descriptor and can handle monotonic illumination variations well in the textured images [29,60,64]. Therefore, we used Local Binary Pattern (LBP) feature [60]. The LBP is an extensively used feature for fields such as face recognition [65], gender [66], ethnicity, and age classification [67]. Timo et al. proposed a method of applying LBP features on the facial recognition problem for the first time and showed better results than several traditional approaches [68]. In [69], the authors used LBP features for removing light effect. Figure 3 shows the example of encoding a LBP feature.

A 3 × 3 sized block is used to obtain a feature value for a point. Each pixel point is encoded as 1 if it is brighter than the center, and as 0 when it is darker than the center. The formula is as follows:(1)$$\begin{array}{c} s\left(x\right)=\left\{\begin{array}{cc} 1, & \mathrm{if}x\geq 0,\\ 0, & \mathrm{if}x<0. \end{array}\right. \end{array}$$
s(x) is converted to a binary number through a difference from the center pixel. Then, the binary code is converted into a decimal number. LBP feature helps reduce computational complexity compared to the original image. In addition, it emphasizes the main texture of face in image. Figure 4 shows each process of feature extraction.

We applied the normalized box filter smoothing before extracting LBP feature to emphasize its result. The LBP feature can be distorted due to noise because it compares brightness value of image pixel by pixel. After 5 × 5 size filtering, we can remove noises simply. This makes the features of the pixels in the concentrated part such as eyes and mouth in image similar. The reason we extract LBP feature from the filtered image is that noise becomes smoother and the feature is more highlighted. Through this process, unnecessary parts of sequence are eliminated and the important feature is emphasized so that the network can extract appearance-based feature more effectively.

#### 3.2.2. Appearance Feature-Based Network Structure

We use 3D convolution for capturing spatiotemporal information. Unlike 2D convolution, 3D convolution captures spatial and temporal information at the same time by using three-dimensional input dataset [70]. To analyze the movement of face when making a facial expression, motion information should be captured from multiple frames. Since 3D convolution has 3D cube formed convolution kernel, it has one more dimension as the time axis. Thus, it preserves the temporal information of the input sequence and makes volume formed outputs.

In this way, feature maps of convolution layer can be connected to contiguous frames so that it can capture motion information. Therefore, spatial and temporal information can be extracted simultaneously by the 3D convolution. As shown in Equation (Equation 2), Shuiwang et al. explained the 3D convolution mathematically [71]. The value at position (x,y,z) on the *j*th feature map in the *i*th layer is given by:(2)$$v_{ij}^{xyz}=tanh\left(b_{ij}+\sum_{m}\sum_{p=0}^{P_{i}-1}\sum_{q=0}^{Q_{i}-1}\sum_{r=0}^{R_{i}-1}w_{ijm}^{pqr}v_{(i-1)m}^{\left(x+p)(y+q)(z+r\right)}\right),$$
where (p,q) is spatial dimension index and *r* is temporal dimension index of kernel, $w_{ijm}^{pqr}$ is the (p,q,r)th value of the kernel connected to the *m*th feature map in the previous layer, and $R_{i}$ is the size of the 3D kernel. tanh() assumes that activation function is the hyperbolic tangent, but other activation function can also be used.

Other layer configurations are similar to those of Yu et al. [72] and Yim et al. [33], who used convolution and pooling layer, respectively. First, 3D convolution layer extracts spatial and temporal features. Every convolution layer uses 5 × 5 × 3 size of kernel and Rectified Linear Unit (ReLU) as activation function [73]. Then, 3D pooling is applied to decrease the number of parameters and cope with the location change of image component. Our pooling layer is max pooling, which delivers only the maximum value among the response results of volume area.

After the max pooling operation, the size of feature map is reduced. Due to 3D pooling [74], dimensional reduction also occurs for the time axis. The maximum value in 2 × 2 × 2 sub-block is mapped to a single pixel in the output 3D feature map. From the max pooling operation, we can get a more robust feature that is the important property for classifying facial expression. The max pooling extracts features regardless of the relative location or direction of important facial parts such as eyes, nose, and mouth.

Every max pooling layer is followed by a batch normalization layer. The batch normalization is one of the ideas that prevent gradient vanishing or gradient exploding [75]. In this paper, we use batch normalization between each layer and drop out algorithm to avoid over-fitting and to increase learning efficiency. Figure 5 shows the detailed configuration of the proposed appearance network.

In Figure 5, “3D Conv.” means 3D convolution layer, “3D Pool.” means 3D Pooling layer, “BA” means the batch normalization layer, and “FC" is a fully connected layer. The numbers under the convolution layers and fully connected layers refer to the number of convolution kernels and classes. #classes means the number of emotion classes and we set it as seven. At the end of our network, softmax function is used for extracting seven emotions as continuous values. The formula of the softmax function for seven emotion labels is as follows:(3)$$s_{i}=\frac{e^{a_{i}}}{\sum_{k=0}^{n-1}e^{a_{k}}},$$
where *n* is the number of emotions to classify and has the same meaning as #classes in Figure 5. $s_{i}$ means the softmax function score of the *i*th class. Softmax is a function that makes total output values as 1 by normalizing the input value between 0 and 1. For softmax function, the output is defined by the number of classes to be classified and the class with the largest output is used as the highest probability.

For training our appearance network, Adam optimizer is used for optimization with learning rate 0.0001 with a default parameter [76]. We use the categorical cross entropy as the loss function for a probabilistic interpretation of the scores [77]. The cross entropy loss function is calculated as follows:(4)$$\begin{array}{c} L=-\sum_{j=0}^{n-1}y_{j}log\left(s_{j}\right), \end{array}$$
where $y_{j}$ is the actual label that is given as one-hot vector and $s_{j}$ is the predicted label for the *j*th element.

### 3.3. Geometric Feature-Based Network

#### 3.3.1. Landmark Detection

Geometric network uses facial landmark coordinate vectors as input to train geometric features of face. For landmark extraction, we adopt real-time landmark extraction technique using the ensemble of regression trees by Kazemi [78]. It takes 68 landmarks from a face by using iBUG 300-W face landmark dataset, which is designed to extract 68 landmarks of a face [79]. Figure 6 shows the extracted landmarks and representation on the surprise labeled face image of three representative databases.

In this paper, we propose a method to contain more information for landmark vectors. For this purpose, we emphasize landmark in areas where movement is larger (larger motion energy) when facial expression changes instead of using 68 landmarks directly. In [30], the author defined 13 important points including eyebrows, eyes, nose, and mouth area. Kim et al. selected 18 points based on the action unit [29]. We calculate the amount of change to identify most activated parts when expression changes using Equation (Equation 5).
(5)$$\begin{array}{c} Landmark_{neu}^{i}=\left[X_{neu},Y_{neu}\right],\\ Landmark_{emo}^{i}=\left[X_{emo},Y_{emo}\right],\\ Landmark_{diff}^{i}=|X_{neu}-X_{emo}\left|*\right|Y_{neu}-Y_{emo}|. \end{array}$$

$Landmark_{neu}^{i}$ is the set of landmark coordinates of *i*th neutral labeled image and $Landmark_{emo}^{i}$ means the same but it is for emotional image. *X* and *Y* are the vectors consisting of 68 landmark points. Each of them has unique number. $Landmark_{diff}^{i}$ is to calculate the variation of landmark coordinates in terms of motion energy by the difference between *X* and *Y* coordinates of image with emotion and neutral. The higher is $Landmark_{diff}^{i}$, the more active are the expression changes. We calculated every $Landmark_{diff}$ from the CK+, MMI, and FERA datasets. Then, we sorted each landmark by $Landmark_{diff}$. They are shown by the color and size of the circles on the face in Figure 7.

The largest red circle means the largest $Landmark_{diff}$ and the smaller is the size of the circle, the smaller is the $Landmark_{diff}$ value. In Figure 7, large movements are mostly captured around the eyes, nose, and mouth. We select the top 13 landmarks among high $Landmark_{diff}^{i}$ value of points as used, e.g, by Liu et al. [30]. Figure 8 is a representation of these 13 landmarks on samples faces of each dataset. Figure 8a is from the CK+ dataset and Figure 8b is from the MMI dataset.

Although this is similar to the main parts used in papers using partial landmarks from faces by Kim et al. [29] and Liu et al. [30], many movements were also detected in the face outline beside the eye and center of the nose. We designed the distribution of points symmetrically in the dense part. The main part with large difference between expressions are included, but the dominant movement of the whole face can be also considered. Figure 9 shows 23 landmark points including 10 newly added points which take into account the symmetry of the extracted 13 landmark points. As shown in Figure 9, the green marked points are newly added points and red marked points are part of the top 13 points obtained through calculation.

Each of the 23 landmarks has x, y coordinates; therefore, we concatenate 23 points of coordinates. As a result, we use 46-length vector as input of geometric network. The input vector is as follows:(6)$$\begin{array}{c} V_{emo}^{i}=\left[x_{emo}^{0}-x_{neu}^{0},y_{emo}^{0}-y_{neu}^{0},\ldots ,x_{emo}^{20}-x_{neu}^{20},y_{emo}^{22}-y_{neu}^{22}\right], \end{array}$$
where $V_{emo}^{i}$ is the landmark difference vector of *i*th frame of emotion *e*. Every $V_{emo}$ is made by three difference frames. Vectors for all frames are concatenated to use an input for geometric network. $x_{emo}$, $y_{emo}$, $x_{neu}$, and $y_{neu}$ are the coordinates of 23 landmarks. The subscript neu means a neutral expression and emo means specific emotion. They are listed in order from left to right, top to bottom, and edge of eyes and lips coordinates are constructed in clockwise order based on the work of Sagonas et al. [79].

In the case of the neutral labeled frame, they are made by just concatenating landmarks extracted from three consecutive neutral frames since we cannot calculate landmark differences for neutral expression. Each frame has 46 × 1 shape of vector. After concatenating three frames, the input vector size becomes 46 × 3.

#### 3.3.2. Geometric Feature-Based Network Structure

Geometric network receives 23 important landmark vectors. Input landmark vectors are made by calculating the difference between neutral face and emotional face. This means the temporal difference, thus we can train with temporal information even based on the 2D convolution using these landmark vectors. Figure 10 shows the flow of making the input vector for geometric network.

In this paper, we construct geometric network modified from the VGGNet structure [80]. The VGGNet is a well-known model that achieves good performance by implementing a simple deep network and won second place in the ImageNet Large Scale Visual Recognition Challenge (ILSVRC) 2014 [81]. Depending on the depth and setting of layers, it is called VGG16, VGG19, etc. Among them, we modified VGG16 network structure. The original VGG16 network has five convolution blocks, which have 2, 2, 3, 3, and 3 convolution layers each. Our input data structure is very simple as 46 × 3 sized of vector. When a network has deep layer structure or large size of filter kernel, it is difficult to apply to simple input structure because the size of input data is reduced quickly. Therefore, we use only the first three block structure of VGG16 network and increase the number of the convolution layers of the second block from 2 to 3. In addition, the shape of input dataset was adjusted to be one-dimensional vector, and then the 3 × 3 filter used in the original VGG16 network was changed to 3 × 1. The detailed structure of geometric feature-based network is shown in Figure 11.

For training geometric network, we use the Adam optimizer with 0.0001 as learning rate with a default parameter and the cross entropy loss function the same as the designed appearance network.

### 3.4. Joint Fusion Classifier

In this Section 3.4, a joint fusion classifier is designed to combine the results of the two networks. We design the joint fusion classifier with three fully connected layers to classify more accurately, as shown in Figure 12. This classifier is used to make the two networks reinforce each other when appearance network or geometric network does not get the correct answer. In other words, joint fusion classifier can compensate the result from each networks to make more accurate result. For convenience, we call the appearance network and geometric network by Network 1 and Network 2. To utilize two networks, we integrate the outputs of two networks using Equation (Equation 7).
(7)$$\begin{array}{c} x_{j}=\alpha o_{j}^{1}+\left(1-\alpha \right)o_{j}^{2}, \end{array}$$
where $x_{j}$ is the input of joint fusion classifier, α is the weight that determined by experiment, and $o_{j}^{1}$ and $o_{j}^{2}$ are *j*th softmax outputs of appearance network and geometric network, respectively. We set α as 0.54 empirically. $x_{j}$ is (*n*, #classes) shaped vector. Originally, the softmax function makes the output probabilities from 0 to 1 and the total sum of the vector to be 1. Since $x_{j}$ is made by weighted sum of the outputs from two networks, the total sum of $x_{j}$ vector is over 1.

Through the joint fusion classifier, the final result from two networks is more accurate. We denote the joint fusion classifier by a sequence of letters and numbers, e.g., I (*n*,7)-FC1 (105)-FC2 (49)-FC3 (7), where *n* means the number of output features of each networks. FC1 and FC2 layers use ReLu and the last FC3 layer uses softmax as the activation function. In addition, the joint fusion classifier uses cross entropy as the loss function and the Adam optimizer. As a result, we can make the final decision via the designed joint fusion function for emotion prediction.

## 4. Experimental Results and Discussion

In this section, we take the experiment and its environment in detail. In addition, we compare with other state-of-the-art algorithms in facial expression recognition using spatiotemporal networks. Thus, we demonstrate that the proposed algorithm shows comparable performance. In following section, we present the experimental results and analyze the performance with some experimental results.

### 4.1. Performance as the Number of Input Frames

This experiment was to determine how many frames is the most appropriate when we connect some continuous frames to create an input dataset structure. The number of minimum frames of databases is about 10 frames. Thus, we took the experiment by concatenating up to seven frames excluding 3–5 frames for neutral labeled frame. In addition, odd number of frame is suitable since the convolution operation should be vertically and horizontally symmetrical except for the center. Therefore, we compared cases of three, five, and seven frames of input data constructed using the CK+, MMI, and FERA datasets. Except for condition for the number of frames, applying LBP feature after normalized box filtering, cropping face area, resizing to 128 × 128, and the network structure was same for experiment.

To construct multiple frames input for the CK+ dataset, we should follow a rule because of the characteristic of this dataset. The neutral sequences in the CK+ dataset are at the beginning of the video. For making three consecutive frames as input, we assigned the first three frames for neutral labeled frames and the next three frames were designated emotional labeled frames. For five frames, similar to when using three frames as input, we used the first five consecutive frames for neutral frames and the other five frames for emotional frames. However, in the case of using seven frames, we connected seven frames differently. We used the first five frames and the fifth frame was used twice to fill the void because, if we use the first seven frames directly, part of the emotion would be included in some cases. For the emotional frames, we used seven frames similarly as with the other cases. Figure 13 shows the examples of the neutral labeled frames extracted from the surprise labeled sequence. Figure 13b–d represents three frames, five frames, and seven frames of neutral label, respectively. Figure 13a shows the frames from the beginning to the seventh.

Some emotional frames are included such as the sixth and seventh frames. Thus, we duplicated the fifth frame to make seven neutral frames in the case of the CK+ dataset. For the MMI dataset, we separated frames for each emotion before making inputs. Because the MMI dataset, unlike the CK+ dataset, has a flow that starts from neutral emotion to the peak of one expression and then returns to the neutral emotion, it is difficult to separate frames with fixed time. Therefore, we manually categorized frames according to the emotion.

On the other hand, the FERA dataset does not have neutral label but has relief label. To match conditions with other databases in our study, we used a relief label as a baseline emotion of whole emotions similar to the neutral label in other datasets. For the MMI and FERA datasets, we took consecutive frames from the beginning according to conditions for the number of frames since they have emotion frames extracted by the number of classes.

After saving the dataset per condition for the number of frames, we experimented following our proposed scheme. Each dataset was trained on the same networks, the appearance network and the geometric network. Then, we analyzed results of the joint fusion classifier which combine the appearance network and the geometric network with 23 landmark points without using the weight. The accuracy in Table 3 represents the average accuracy of 10 trials for different number of frames. The experiment results show that the accuracy is the highest when using the input consisting of three frames. Thus, we set three consecutive frames as input data in all experiments.

### 4.2. Performance as the Number of Landmarks

We designed a new method to select 23 landmarks partially that represent the main parts of face based on the proposed method and related work from the other paper [30]. Originally, we can extract 68 landmarks from the whole face through the landmark extraction method [77]. We constructed three datasets to three of 46-length vectors from the coordinates of 23 landmarks, as shown in Equation (Equation 6). Figure 14 shows the 10 trial results of the geometric network on the CK+ database.

The average accuracy of using 68 landmarks was 95.72% and the average accuracy of using only 23 landmarks was 95.68%. The accuracy of 68 landmarks was slightly higher. However, their overall performance is very similar. Using only 23 landmarks can reduce the capacity of the input data by about 1.6 times compared to 68 landmarks. As the number of input data increases, landmark extraction process takes a long time. Therefore, we can expect the same effect as considering the entire landmarks on the face while using only 23 landmarks.

### 4.3. Optimal Weight Analysis for Joint Fusion Classifier

We demonstrated which input structure of the joint fusion classifier is more effective to increase the performance of classification with two different ways. The first way is to concatenate outputs of two networks and the second way is taking element-wise operation. The weighted fusion method is represented as Equation (Equation 7) in Section 3.4. As discussed in the previous section, α is determined by experiment to find the value that shows the highest accuracy. α has values with a difference of 0.033 from 0.6 to 0.4. This is a range of values to make the impact of two networks as similar as possible. We assigned different weights on the output of each network. The average accuracy of weighted sum method is higher than concatenation method for all datasets. This is because concatenation method does not result in complementary interactions between the two networks. The concatenation method is even less accurate than using single network. Therefore, Table 4 only shows the average accuracy of the weighted fusion method. The highest accuracy in each dataset is emphasized in bold.

To find the optimal weight factor, we took experiments with various weight values by shifting. The accuracy was higher when adding the larger weight on the appearance network for all datasets. Weighting on the poor network does not mean a decrease of performance. Since the joint fusion classifier is a complementing learning process for classification, we can observe that the accuracy does not drop significantly in the case of adding a larger weight on the geometric feature-based network.

As a result, we found that the average accuracy is highest when α is 0.54. From this result, we set 0.54 as α for the weighted fusion method in all experiments.

### 4.4. Performance of the Accuracy

We demonstrate that the proposed scheme shows a competitive performance compared with state-of-the-art methods. Among various techniques for facial expression recognition, we compared with spatiotemporal neural network approaches or hybrid network approaches. Table 5 shows input construction and model setting of state-of-the-art methods which were compared with the proposed method. For experiments, we used three datasets: CK+, MMI, and FERA. The number of image sequences in each dataset is listed in Section 3.1. In our networks, we used three frames as input and weights to make an input for the joint fusion classifier, as mentioned in the previous subsection. We took 10 trials to measure the accuracy.

The compared accuracy of each method using the CK+ dataset is shown in Table 6. Table 7 shows the comparison of experimental results using the MMI dataset. In Table 6, Table 7 and Table 8, the bold faces are the accuracy values of the proposed appearance feature-based network (Proposed (App.)), geometric feature-based network (Proposed (Geo.)), and the proposed deep joint spatiotemporal network (Proposed (Joint)), respectively. In Table 7, result of the geometric network is not the best result. It is even less than STFR model [83] and 3DIR model [11]. However, we can see that the result of the designed joint fusion classifier outperforms compared all of other methods. This is because, even if the results of either network are not good, the correct answer of the appearance network affects not correct network through the proposed joint fusion classifier.

For the FERA dataset, we took experiment with the same hybrid network but different number of output class. Because the FERA dataset has only five emotion labels unlike seven labels in other datasets, as mentioned in previous section, we used relief label as neutral label. Table 8 shows the comparison of experimental results. Comparing to other methods, our proposed method outperforms in the FERA dataset, especially. This is because many dynamic motions and fast temporal movements in the FERA dataset were well extracted and trained through our networks.

Table 9, Table 10 and Table 11 show the confusion matrices of the proposed deep joint spatiotemporal network (DJSTN) on three datasets. Each value means recall to evaluate the model. The bold face denotes the recognition accuracy value for a given expression in Table 9, Table 10 and Table 11. In Table 9, high recognition rates near perfection are shown in the CK+ dataset. Table 10 shows that the highest recognition rate in the MMI dataset is for surprise emotion. On the other hand, recognition rates of angry, disgust, fear, and sad emotions are relatively low. The result of the FERA dataset is shown in Table 11. As mentioned above, the FERA dataset has five emotion labels. There are some confusions between emotions except sadness. Despite a slightly lower recognition rate of anger emotion, the proposed scheme achieves outperforming result in the FERA dataset.

The proposed scheme needs the neutral frame to be used for the reference image in the geometric feature-based network. In real situations and applications, it can be very difficult to give a neutral frame for providing the natural emotion in prior. To solve this situation, the proposed scheme needs to capture video segment (neutral frames) with user’s neutral expression for 1–2 s, in real application. In addition, we focused on the proposed deep joint spatiotemporal network (DJSTN) to improve the recognition accuracy, not on implementation efficiency. If we are interested in the computational efficiency, the 2D convolution should be carried out via the fast Fourier transform (FFT) operation [84,85].

## 5. Conclusions

We have proposed an efficient deep joint spatiotemporal network (DJSTN) that combined the appearance network and geometric network with joint fusion classifier. To overcome problem of the traditional 2D CNN model that only was trained with spatial features, the proposed appearance network used 3D convolution and multiple input frames with the normalized LBP feature to extract spatial and temporal features at the same time. For the geometric network, we selected dominant 23 facial landmarks that could represent the whole facial landmarks through the experiment that measured the movement of landmarks when facial expression changes. The geometric network used the difference of landmark coordinates between neutral face and emotional face as input so that it trained temporal information with 2D convolution. As a result, we combined outputs of these two networks as the joint fusion classifier. The designed joint fusion classifier closely classified as the correct answer from the weighted fusion technique of two networks.

All experiments were performed with three frames of input, and we demonstrated that our facial expression recognition algorithm achieved more accurate results than state-of-the-art methods. In addition, we have shown that the proposed deep joint spatiotemporal network (DJSTN) reinforced the performance of the single network by 2.7%, 10.2%, and 6% on average accuracy for the CK+, MMI, and GEMEP-FERA datasets by combining two networks with the designed joint fusion classifier. It means that the joint fusion classifier played a role to improve the performance of two networks more robustly. The overall proposed algorithm outperformed the existing methods with 99.21% of the accuracy for the CK+ dataset, 87.88% for the MMI dataset, and 91.83% for the GEMEP-FERA dataset, respectively.

## Figures and Tables

**Figure 1 sensors-20-01936-f001:**
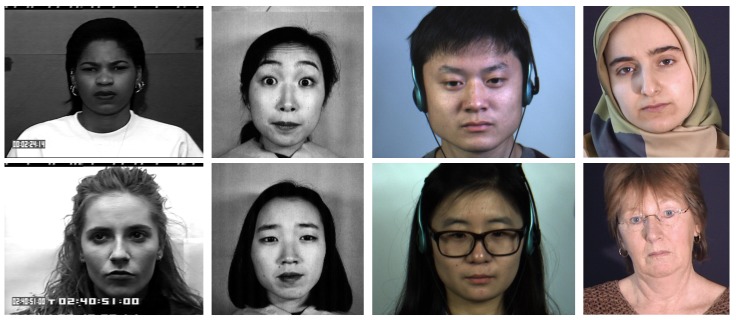
Datasets with different genders, races, and ages.

**Figure 2 sensors-20-01936-f002:**
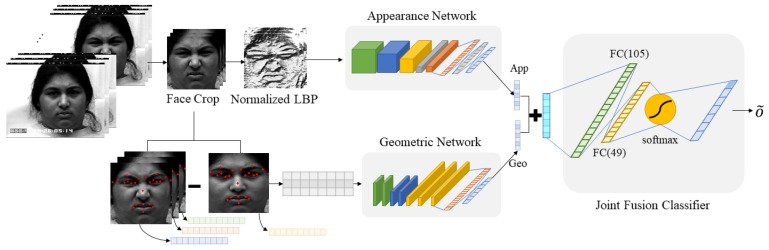
The overall structure of the proposed facial expression recognition scheme.

**Figure 3 sensors-20-01936-f003:**
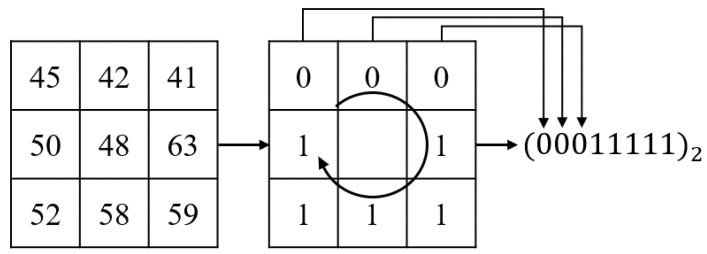
Example of encoding a LBP feature.

**Figure 4 sensors-20-01936-f004:**
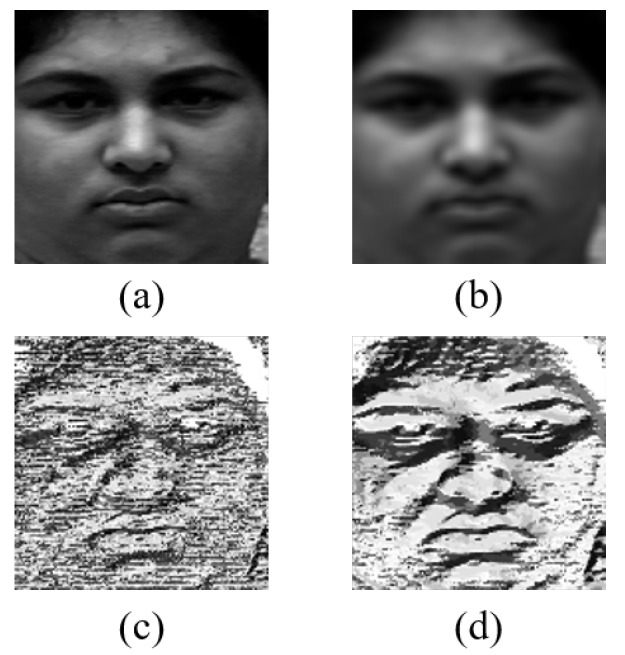
Feature extraction: (**a**) original image; (**b**) filtered image; (**c**) LBP feature without filtering; and (**d**) LBP feature with filtering.

**Figure 5 sensors-20-01936-f005:**
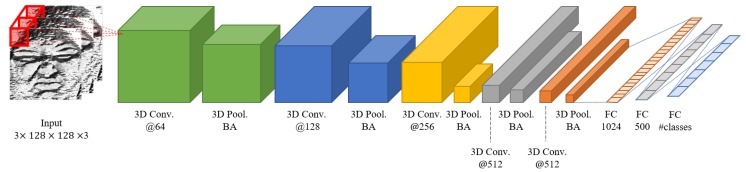
Appearance feature-based spatiotemporal network.

**Figure 6 sensors-20-01936-f006:**
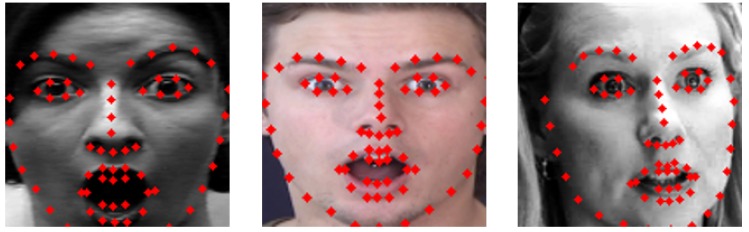
Examples of 68 landmarks detection: (**Left**) CK+; (**Middle**) MMI; and (**Right**) AFEW.

**Figure 7 sensors-20-01936-f007:**
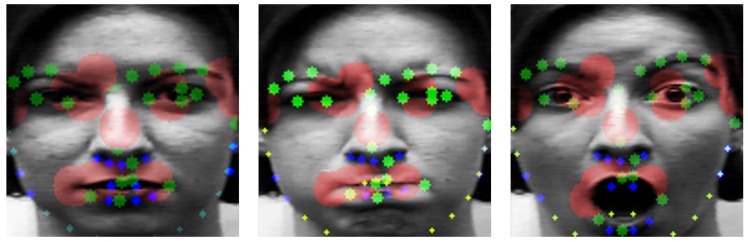
Movement distribution of landmarks.

**Figure 8 sensors-20-01936-f008:**
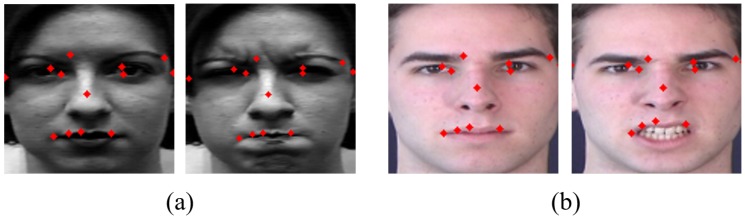
Top 13 landmarks: (**a**) CK+ dataset; and (**b**) MMI dataset.

**Figure 9 sensors-20-01936-f009:**
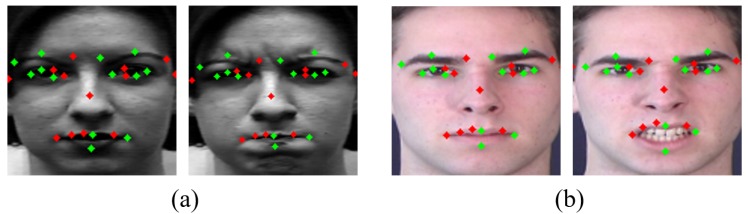
Twenty-three landmarks used as the input of geometric network: (**a**) CK+ dataset; and (**b**) MMI dataset.

**Figure 10 sensors-20-01936-f010:**
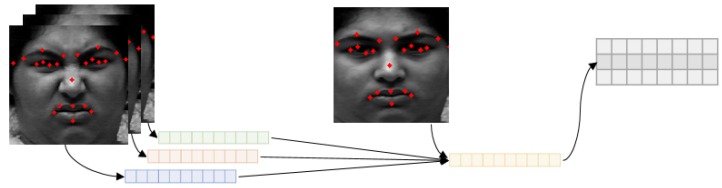
The input vector for the geometric feature-based network.

**Figure 11 sensors-20-01936-f011:**
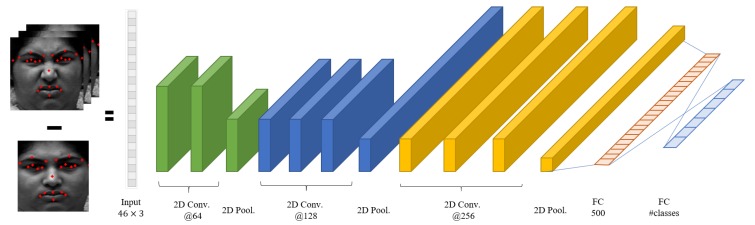
Geometric feature-based network.

**Figure 12 sensors-20-01936-f012:**
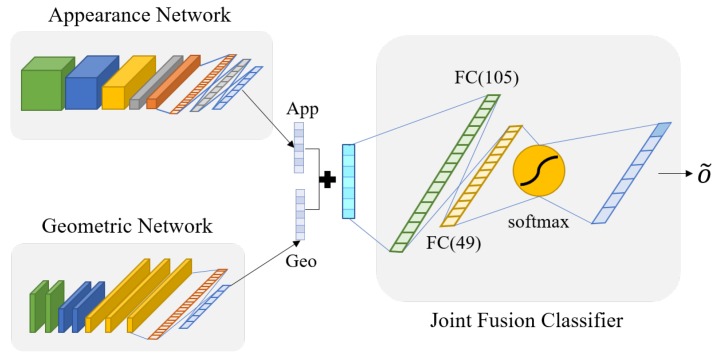
The structure of joint fusion classifier.

**Figure 13 sensors-20-01936-f013:**
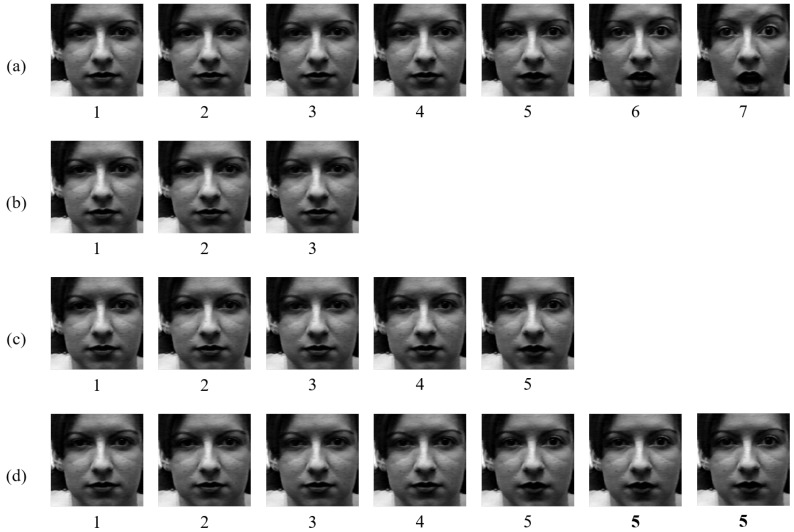
The examples of the neutral labeled frames: (**a**) seven consecutive frames; (**b**) neutral frames for three; (**c**) neutral frames for five; and (**d**) neutral frames for seven.

**Figure 14 sensors-20-01936-f014:**
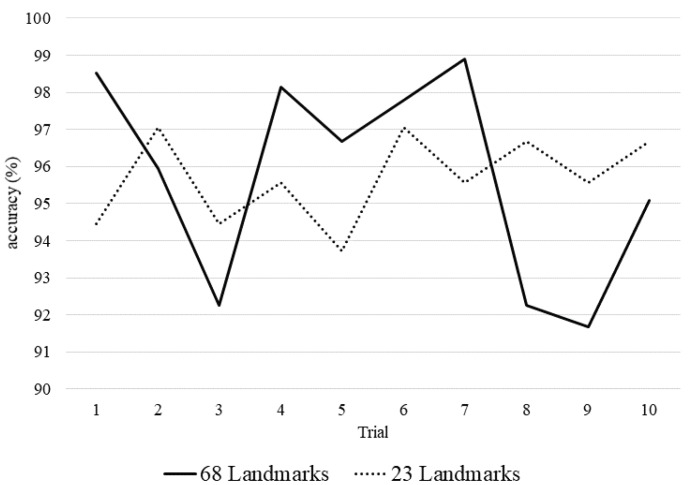
Comparison of accuracy according to the number of landmarks.

**Table 1 sensors-20-01936-t001:** Analysis of the existing FER systems for business applications.

Items	Description	Developers
Project Oxford [41]	As part of the Machine Learning (Machine Learning) project, it provides API including Face API for machine learning technology, which is being promoted to four major categories.	Oxford Univ. and Microsoft
Face Reader [42]	The Project Analysis Module includes the Stimulus Presentation Tool. By designing a test, this tool automatically shows the test participants the stimuli while FaceReader accurately analyzes the participant’s face.	Noldus
Emotient [43]	Emotient is spearheading the use of machine learning for facial expression analysis, a new category of natural user interface that is rapidly expanding as more companies seek to improve technology responsiveness and increase customer engagement.	crunchbase
Affectiva [44]	The first multi-modal in-cabin sensing AI that identifies, from face and voice, complex and nuanced emotional and cognitive states of drivers and passengers. This helps improve road safety and the transportation experience.	Affectiva
EmoVu [45]	EmoVu specializes in creating emotionally intelligent tools that can perceive the emotions of people by analyzing microexpressions using webcams.	Eyeris
Kairos [46]	Kairos provides featured platforms such as face detection/recognition, age, gender detection, etc.	Kairos
Nviso [47]	Nviso develops face detection, face verification, demographic detection, emotion classification, action unit detection, and pose detection.	Nviso
Sightcorp [48]	Sightcorp provides technologies for face recognition, emotion detection, attention analysis, and mood detection.	Sightcorp
SkyBiometry [49]	SkyBiometry is state-of-the-art face recognition and face detection cloud biometrics API allowing developers and marketers.	SkyBiometry
Face++ [50]	It ensures that operator behind a transaction is a live human by facial landmarks localization, face tracking technique, etc.	Face++
Imotions [51]	Motions helps you quantify engagement and emotional responses. The iMotions Platform is an emotion recognition software that seamlessly integrates multiple sensors.	Imotions
CrowdEmotion [52]	An emotion inspired artificial intelligence company that enables technology to see, hear, and feel the way humans do.	CrowdEmotion
FacioMetrics [53]	FacioMetrics develops facial analysis software for mobile applications including facial image analysis—with all kinds of applications including augmented/virtual reality, animation, and audience reaction measurement.	Facebook Research
Findface [54]	A face recognition technology developed by the Russian company NtechLab that specializes in neural network tools. It compares photos to profile pictures on social network Vkontakte and works out identities with 70% reliability	Findface

**Table 2 sensors-20-01936-t002:** The number of input images as emotions and datasets.

	Neu	Ang	Dis	Fea	Hap	Sad	Sur	Total
CK+	316	360	448	192	552	224	624	2716
MMI	366	288	492	258	204	366	366	2340
FERA	603	529	-	479	606	695	-	2915

**Table 3 sensors-20-01936-t003:** The recognition accuracy according to three different input data structures (%).

	3 Frames	5 Frames	7 Frames
**CK+**	99.27	94.45	95.57
**MMI**	87.23	83.76	81.88
**FERA**	90.67	87.85	87.15
**Average**	**92.39**	**88.68**	**88.20**

**Table 4 sensors-20-01936-t004:** Comparison of the average accuracy according to α.

α	CK+	MMI	FERA
**0.60**	98.87	87.28	91.31
**0.57**	99.20	86.64	91.78
**0.54**	99.21	**87.88**	**91.83**
**0.50**	**99.36**	87.49	91.79
**0.47**	99.09	87.51	90.77
**0.44**	99.02	87.23	90.60
**0.40**	99.07	86.79	89.13

**Table 5 sensors-20-01936-t005:** Analysis of the state-of-the-art methods.

Method	Database	Input Construction	Model
DTAGN [33]	CK+	DTAN, DTGN	Hybrid network
3DIR [11]	CK+, MMI, FERA	Multiple frames, Facial landmarks	3D CNN, LSTM, Inception-ResNet
DESTN [82]	CK+	Single image frame, Multiple landmarks	Hybrid network
nestedLSTM [72]	CK+, MMI	Multiple frames	3D CNN, LSTM
STCNN-CRF [32]	CK+, MMI, FERA	Multiple frames	2D CNN, CRF, Inception-ResNet
STFR [83]	MMI	Multiple frames	LSTM

**Table 6 sensors-20-01936-t006:** The performance comparison of the recognition accuracy in the CK+ dataset (%).

Methods	Accuracy (%)
DTAGN [33]	97.25
STCNN-CRF [32]	93.04
3DIR (S/I) [11]	93.21
DESTN [82]	98.50
nestedLSTM [72]	99.80
Proposed (App.)	**97.50**
Proposed (Geo,)	**95.68**
Proposed (Joint)	**99.21**

**Table 7 sensors-20-01936-t007:** The performance comparison of the recognition accuracy in the MMI dataset (%).

Methods	Accuracy (%)
STCNN-CRF [32]	68.51
3DIR [11]	77.50
STFR [83]	78.61
nestedLSTM [72]	84.53
Proposed (App.)	**84.84**
Proposed (Geo.)	**75.24**
Proposed (Joint)	**87.88**

**Table 8 sensors-20-01936-t008:** The performance comparison of the recognition accuracy in the FERA dataset (%).

Methods	Accuracy
STCNN-CRF [32]	66.66
3DIR (S/I) [11]	77.42
Proposed (App.)	**87.80**
Proposed (Geo.)	**85.48**
Proposed (Joint)	**91.83**

**Table 9 sensors-20-01936-t009:** Confusion matrix of joint fusion classifier in the CK+ dataset. NE, Neutral; AN, Angry; DI, Disgust; FE, Fear; HA, Happy; SA, Sad; SU, Surprise.

Actual values		NE	AN	DI	FE	HA	SA	SU
	NE	**0.97**	0.00	0.00	0.00	0.00	0.03	0.00
	AN	0.00	**0.99**	0.00	0.00	0.00	0.01	0.00
	DI	0.00	0.01	**0.99**	0.00	0.00	0.00	0.00
	FE	0.00	0.00	0.00	**0.98**	0.02	0.00	0.00
	HA	0.00	0.00	0.00	0.00	**1.00**	0.00	0.00
	SA	0.00	0.00	0.00	0.02	0.00	**0.98**	0.00
	SU	0.01	0.00	0.00	0.00	0.01	0.00	**0.98**
	Predicted values							

**Table 10 sensors-20-01936-t010:** Confusion matrix of joint fusion classifier in the MMI dataset. NE, Neutral; AN, Angry; DI, Disgust; FE, Fear; HA, Happy; SA, Sad; SU, Surprise.

Actual values		NE	AN	DI	FE	HA	SA	SU
	NE	**0.89**	0.00	0.03	0.02	0.01	0.03	0.03
	AN	0.16	**0.62**	0.10	0.01	0.01	0.08	0.01
	DI	0.09	0.09	**0.79**	0.01	0.02	0.01	0.00
	FE	0.07	0.00	0.10	**0.78**	0.00	0.05	0.10
	HA	0.15	0.00	0.00	0.00	**0.85**	0.00	0.00
	SA	0.10	0.06	0.00	0.06	0.02	**0.76**	0.00
	SU	0.05	0.00	0.00	0.02	0.02	0.00	**0.91**
	Predicted values							

**Table 11 sensors-20-01936-t011:** Confusion matrix of joint fusion classifier in the FERA dataset. RE, Relief; AN, Anger; FE, Fear; JO, joy; SA, Sadness.

Actual values		RE	AN	FE	JO	SA
	RE	**0.90**	0.03	0.01	0.06	0.00
	AN	0.15	**0.83**	0.00	0.02	0.00
	FE	0.04	0.01	**0.96**	0.03	0.00
	JO	0.05	0.02	0.01	**0.92**	0.00
	SA	0.01	0.00	0.01	0.00	**0.98**
	Predicted values					

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
