# Peer review of "Deep Joint Spatiotemporal Network (DJSTN) for Efficient Facial Expression Recognition"

_sensors, 2020, doi:10.3390/s20071936_

Round 1
Reviewer 1 Report
The authors propose a facial expression recognition method based on 3DCNN. The main idea is to employ the 3D CNN to extract the spatial- temporal information and employ the traditional 2D CNN to explore the geometric information. These two kinds of features are combined as the input of the last three fully connected layers. Although better results are reported, there are some issues remain to be further addressed.
- The contribution (novelty) is limited. The combination of these two kinds of features is straightforward. The authors train their network based on three datasets. Therefore, it is reasonable to achieve better performance than the method using only one dataset (e.g., [59] and [61]).
- In line 80-82, why they cannot improve the accuracy?
- In line 135, why do you need to make the number of samples in each dataset similar?
- I think the authors spend too much space on introducing the details of the network structure, such as the 3D convolution. The 3D CNN was proposed for activation recognition 7 years ago (at least). It is not necessary to mention these details. Figure 2 and Figure 12 are quite similar. You can consider to remove Figure 5, Figure 6, Figure 11 and Figure 12.
- How to detect the natural emotion from the video? I noticed that the geometric features are based on the difference between given frames and the natural frame. It would be much more interesting if the model can automatically recognize the emotion without the knowledge about the natural emotion.
- A lot of orthographic and grammatical errors have been encountered in the manuscript, not limited to the following ones.
* In academic English, people always use ‘therefore’ rather than ‘so’ at the beginning of sentences. (line 67, line 135 and many more)
* line 78: ‘Using the expanded features were used to recognize on 3D face database.’
* line 87-89: ’ Using secondary …were considered the way…’.
* line 96: ’ supporting’
* line 122: ‘though’
* line 127: ‘Also, It….’
* the difference between ‘data’, ‘dataset’ and ‘database’
* line 165: ‘MTCNN (full name???)’
Author Response
The authors would like to express their sincere thanks to the reviewers for their good comments. Could you refer to the attached replies for comments?

Reviewer 2 Report
In this work, the authors propose an efficient deep joint spatio-temporal features for facial expression recognition based on the deep appearance and geometric neural networks. For the geometric network, dominant 23 facial landmarks are selected to express the movement of facial muscle through the analysis of energy distribution of whole facial landmarks. Based on these features, we combine them by the designed joint fusion classifier to complement each other. In general, the manuscript is interesting and well writhed there are some issues which need to be clarify:
Major revisions:
- Related works section: I think that some important contributions such as, the Viola-Jones-based facial expression recognition or based on cross modal data association are missing. More discussion could be a nice complement for the current manuscript.
- The main contribution of this work “the Deep Joint Spatio-Temporal” has a little bit lack of novelty. The use of LBP and a 2D convolution are not enough to be the basis of a research manuscript. Why LBP and no other binary feature such as BRISK, BRIEF, ORB were used? I f the authors are interested in the computational efficiency, why the 2D convolution is not carried out via the FFT (Fast Furrier Transform), several works have demonstrated that this is the best way for CNN processing. In my opinion, more discussion about the current formulation, its real performance and its limitations are needed.
Minor revisions:
- There are a little grammatical/style error. In my opinion, a grammar/style revision has to be carried out before the manuscript can be considered for publication.
- There are some typos, as instance (full name???) in page 5, line 165. I recommend to carried out a detailed language revision.
Author Response
The authors would like to express their sincere thanks to the reviewers for their good comments. Could you refer to the attached replies for comments? Thank you.

Reviewer 3 Report
This manuscript contains interesting work from the deep joint spatio-temporal network for efficient facial expression recognition. However, the manuscript is written in style more like a report rather than a research article. The global innovativeness in research hasn't been presented. There are a lot of similar systems (Project Oxford by Microsoft, Face Reader by Noldus, Emotient, Affectiva, EmoVu, Kairos, Nviso, Sightcorp, SkyBiometry, Face++, Imotions, CrowdEmotion, FacioMetrics, Findface). What is your system better? Some figures and tables which involve world-wide novel research should be described and discussed with more details. Please use the newest (2017-2020) Web of Science journal papers.
Author Response

(The authors gave the same response as above.)

Round 2
Reviewer 1 Report
All of the questions and comments have been responded completely and clearly.
Reviewer 3 Report
Please add new functions for your Deep Joint Spatio-Temporal Network in the future.